# Utilising Pancreatic Exocrine Insufficiency in the Detection of Resectable Pancreatic Ductal Adenocarcinoma

**DOI:** 10.3390/cancers15245756

**Published:** 2023-12-08

**Authors:** Declan McDonnell, Paul R. Afolabi, Sam Wilding, Gareth O. Griffiths, Jonathan R. Swann, Christopher D. Byrne, Zaed Z. Hamady

**Affiliations:** 1Human Development & Health, University of Southampton, Southampton SO16 6YD, UK; p.r.afolabi@soton.ac.uk (P.R.A.); z.hamady@soton.ac.uk (Z.Z.H.); 2University Hospital Southampton NHS Foundation Trust, Southampton SO16 6YD, UK; 3Cancer Research UK Southampton Clinical Trials Unit, University of Southampton, Southampton SO17 1BJ, UK

**Keywords:** pancreatic, adenocarcinoma, PDAC, exocrine, insufficiency, PEI, ^13^C, triglyceride, elastase

## Abstract

**Simple Summary:**

The early detection of pancreatic cancer is critical as it is usually too late for potentially curative surgical resection when obvious symptoms such as jaundice have developed. The pancreas has two main functions in the body, namely, an endocrine role, where it produces insulin to control blood glucose levels; and an exocrine role, where it produces enzymes that aid digestion. Impairment of this latter role is known as pancreatic exocrine insufficiency (PEI). It is an established feature of advanced pancreatic cancer, but it is unclear whether it is present in the early stages when surgery may still be an option. This study used two validated methods of testing pancreatic exocrine function, a breath test and a stool test, to demonstrate that exocrine insufficiency is associated with resectable pancreatic cancer.

**Abstract:**

Pancreatic ductal adenocarcinoma (PDAC) is usually diagnosed late, leading to a high mortality rate. Early detection facilitates better treatment options. The aim of this UK-based case–control study was to determine whether two validated tests for pancreatic exocrine insufficiency (PEI), namely, the ^13^C-mixed triglyceride breath test (^13^C-MTGBT) and a faecal elastase (FE-1) test, can discriminate between patients with resectable PDAC versus healthy volunteers (HVs) along with a comparison group with chronic pancreatitis (CP). Discrimination between disease states and HVs was tested with receiver operator characteristic (ROC) curves. In total, 59 participants (23 PDAC (16 men), 24 HVs (13 men) and 12 CP (10 men)) were recruited, with a similar age in each population, and a combined median (IQR) age of 66 (57–71). The areas under the ROC curve for discriminating between PDAC and HVs were 0.83 (95% CI: 0.70–0.96) for the ^13^C-MTGBT, and 0.85 (95% CI: 0.75–0.95) for the FE-1 test. These were similar to CP vs. HV. In conclusion, PEI occurs in resectable PDAC to a similar extent as in CP; further large-scale, prospective studies using these tests in the primary care setting on high-risk groups are warranted.

## 1. Introduction

In 2016, the estimated global incidence and mortality rates for pancreatic ductal adenocarcinoma (PDAC) were 8.14 cases (95% CI: 6.63–9.98) per 100,000 person-years, and 6.92 deaths (95% CI: 3.72–12.89) per 100,000 person years, respectively [1]. The low survival rates reflect the insidious nature of the disease, as it is usually asymptomatic in the earlier stages, and most cases are diagnosed when it is too late for surgical resection [2]. Early detection of PDAC could improve long-term survival, because it has a better prognosis if treated at an earlier stage [3]. Pancreatic exocrine insufficiency (PEI) can manifest due to the loss of digestive enzymes such as lipase and protease from the exocrine pancreas, such that there is insufficient normal digestion [4]. This is a consequence of advanced PDAC, thought to be due to occlusion of the duct by the tumour [5]. Approximately 50% of patients with PDAC report steatorrhea in the few months prior to diagnosis [6]. 

One strategy to optimise earlier diagnosis involves looking at the association between resectable PDAC and PEI to determine whether the latter is a potential biomarker. There are a variety of methods to detect PEI, including a ^13^C-mixed triglyceride breath test (^13^C-MTGBT), which measures the rate of production of ^13^CO_2_ and is reduced if lipase production is impaired; and faecal elastase (FE-1) [7]. Both tests have some utility in the diagnosis of PEI [8]. These tests are less cumbersome than the gold-standard coefficient of fat absorption test, and produce similar results [9]. Most of the studies using the ^13^C-MTGBT have been conducted over a 360-min window [10]; however, it remains a valid test in as little as 240 min, with a cumulative percentage dose recovered of ^13^CO_2_ of 18.9% accepted as the cutoff for establishing a diagnosis of PEI [11]. The ^13^C-MTGBT has been used to establish PEI in unresectable PDAC [12], and to demonstrate an improvement in exocrine pancreatic function following decompression of the biliary system in obstructions caused by inoperable PDAC [13], although it has not been used to evaluate whether PEI is a feature of resectable PDAC. An alternative breath test using ^13^C-trioctanion has been used to establish PEI in those about to undergo pancreatic resection [14], but the ^13^C-MTGBT is considered a better test than the ^13^C-trioctanion test due to the way it is hydrolysed by pancreatic lipase [15]. The definition of PEI using FE-1 is <200 µg/g [8,16]; a study by Matsumoto et al. found PEI to be present in 21/31 (67.7%) participants with PDAC prior to surgical resection [17]. It is unclear whether the ^13^C-MTGBT can be used to detect PEI due to early-stage PDAC, and how it compares to the more commonly used FE-1 test. In events where PEI has been detected in a study population, it is important to note that the most common cause is chronic pancreatitis (CP) [18]. In 2016, the estimated global index and mortality rates for CP were 9.62 cases (95% CI: 7.86–11.78) per 100,000 person-years and 0.09 deaths (95% CI: 0.02–0.47) per 100,000 person-years, respectively [1]. The incidence of CP is only slightly higher than PDAC, but is much more prevalent due to the significantly lower mortality rate. Therefore, any test that detects PEI should then prompt further evaluation to distinguish between PDAC and CP.

The primary aim of this study was to establish whether two validated tests for PEI can discriminate between patients with resectable PDAC and healthy volunteers (HVs). We also studied participants with established CP as a comparison group. This study is the first to use the ^13^C-MTGBT to determine if PEI is a potential biomarker as a feature of early-stage PDAC. New-onset diabetes mellitus (NODM) is also feature of PDAC, and is considered to be the period of time within 36 months from the diabetes diagnosis [19]. It may also manifest as a result of CP due to the loss of pancreatic parenchyma. A secondary aim of this study was to assess the impact of NODM on the PEI test results.

## 2. Materials and Methods

### 2.1. Study Design 

This was a single-site, unblinded, case–control pilot study in the UK between patients with resectable PDAC, CP, and HV. The protocol for this study has been published as part of the DEPEND project, funded by Cancer Research UK [20]. Ethical approval was granted by the Health Research Authority (HRA) and Health and Care Research Wales (HCRW) to be conducted in (and sponsored by) the University Hospital Southampton NHS Foundation Trust (UHSFT) (IRAS project ID: 286297, REC reference 20/NS/0105, on 21 October 2020) with study design and statistical support from the Cancer Research UK Southampton Clinical Trials Unit.

### 2.2. Participants

Male and female participants aged from 30 to 85 years old within the UK catchment areas of Hampshire, Dorset, the Isle of Wight, and the Channel Islands were invited to participate. Patients with resectable PDAC were recruited in person between the pre-operative stage and after having their diagnosis discussed at UHSFT. They were unable to participate if the disease was deemed inoperable after the regional multi-disciplinary team meeting. Participants with existing CP were approached after routine clinic appointments at UHSFT. HVs were recruited from a pool of established volunteers via the Project Management Recruitment Team based at UHSFT, or willing friends or family members of the PDAC participants who were keen to be involved in the trial. Each individual was given an appropriate participant information sheet explaining the research; they then signed an informed consent form.

### 2.3. Interventions

The study was conducted in the National Institute for Health and Care Research (NIHR) Southampton Clinical Research Facility (CRF) at UHSFT. Participants were required to fast overnight for at least 12 h prior to undergoing the ^13^C-MTGBT. If they were taking pancreatic enzyme replacement therapy (PERT), they were asked to refrain from taking their dose on the morning of the study. They were asked to provide spot stool samples for FE-1 concentrations by using a commercially available ELISA kit or to take the kit home and deliver it to the hospital or their local GP practice within the next seven days. Each participant provided two baseline breath samples by blowing into 2 × 10 mL exetainer tubes through a straw while wearing a facemask. They were then given an oral dose of 250 mg of 2-[^13^C]-octanoyl-1,3-distearin (^13^C-MTG-Cambridge Isotope Laboratories, Andover, MA, USA) together with a solid test meal consisting of a crispbread or other gluten-free alternative, 10 g of fat in the form of butter or dairy-free alternative, and 200 mL of water. Postprandial breath samples were then collected every 30 min for 240 min, as this has been demonstrated to be an adequate timeframe to collect the samples [11]. No food or drinks except for still water were allowed during the duration of the breath test study. No adverse effects from consuming the test meal were recorded by any participant. The collected breath samples were analysed to calculate the ratio of ^13^CO_2_ to ^12^CO_2_ using a Continuous Flow Isotope Ratio Mass Spectrometer (SERCON Ltd, Crewe, UK) at the NIHR Southampton Biomedical Research Centre Mass Spectrometry Laboratory. The results of the ^13^C-MTGBT were then expressed as the cumulative percentage dose recovered of ^13^CO_2_ over 240 min (cPDR_240_). A detailed explanation for how this is calculated is provided in Appendix A.

### 2.4. Sample Size Considerations

The sample size calculation for the proposed study was based on a previous investigation assessing pancreatic exocrine function using a ^13^C breath test in healthy subjects and patients with a localised pancreatic mass [14]. The results of the study showed a mean (S.D.) of the recovery of ^13^CO_2_ over 3 h after undergoing the ^13^C-trioctanoin breath test of 42% (S.D. 3.4%) for the healthy controls and 24.2% (S.D. 10.5) for patients with a localised pancreatic mass, with an effect size or mean difference between both groups of 17.8% and a within-group S.D. of 9.1% (based on the S.D. estimates of 3.4% and 10.5% for each group, respectively). It was determined that 25 participants with PDAC and 25 HVs were required to give >99% power to detect a difference of 17.8% between patients with PDAC and the HV group, assuming an SD of 10.5% in patients with PDAC at a significance level of 5% using a two-tailed test. The power calculation was carried out using IBM SPSS SamplePower V.3.

### 2.5. Statistical Methods 

The skewness–kurtosis method was used to test for normality. Participant characteristics were reported as means (with standard deviation) for data that exhibited a normal distribution, and medians (with interquartile range) for data that did not exhibit a normal distribution. Differences in cPDR_240_ and FE-1 results between the three groups were assessed using the Kruskal–Wallis test, as the data did not exhibit a normal distribution. When there was a statistically significant difference detected between the groups, i.e., *p* < 0.05, two-sided post hoc analysis was undertaken using pair-wise comparisons with Bonferroni adjustment. The presence of diabetes mellitus between the different groups was assessed using Fisher’s exact test. Discrimination between disease states and healthy volunteers was tested with receiver operator characteristic (ROC) curves, and the optimal balance of sensitivity and specificity of the ^13^C-MTGBT and FE-1 test was calculated using the Youden index. All other statistical analyses were carried out using Stata 16 (StataCorp, College Station, TX, USA). The data obtained from the ^13^C-MTGBT were processed on the CF-IRMS using SERCON ABCA2 software version 10.0.39 (07) (SERCON Ltd, Crewe, UK) to calculate the cPDR_240_. The two sets of data at each time point were combined and divided by two to achieve an average percentage. A copy of the statistical analysis plan is available as Appendix A. The STROBE cohort reporting guidelines were followed throughout.

## 3. Results

### 3.1. Participant Demographics

Sixty participants were invited and participated in the initial recruitment stage between 16 March 2021 and 3 January 2022: suspected PDAC (25), HVs (25), and CP (10). Four of those with suspected PDAC were found to have alternative pathology on histology (cholangiocarcinoma (*n* = 2), neuroendocrine neoplasm (*n* = 1), and chronic pancreatitis (*n* = 1). Two had undergone recent endoscopic stenting inserted between recruitment and attending the CRF, and another had more advanced PDAC than initially thought; thus, these were all discounted from the final analysis (*n* = 7). One of the participants with CP declined to participate after 90 min of breath sample collection, so they were excluded from the analysis. To try and compensate for these losses, a further six participants with suspected PDAC (one of whom was later found to have cholangiocarcinoma on histology) and one with CP were recruited between 4 January and 16 March 2022. 

By the end of the recruitment period, there were 23 participants with resectable PDAC. Of these, there were 19 with disease in the head of the pancreas (HOP-PDAC), and four with distal disease in the body or tail. Only those participants who had PDAC confirmed by histology, either following surgical or endoscopic intervention, were included for analysis. The median age was 65 (IQR 56–75) and 16 (69.6%) were male (Table 1). One male participant (aged 62) who was recruited as a PDAC participant was found to have features consistent with an inflammatory mass secondary to CP rather than malignancy on histology; therefore, they were transferred to the CP group. There were 24 participants in the HV group with a median age of 63 (IQR 58–71) and 13 (54.2%) were male. There were originally 25 in this group, but 1 male participant (aged 64) in this group was found to have significantly lower cPDR_240_ (6.78%) than anticipated, which necessitated further investigations as part of the ethical approval for the study. Following endoscopic ultrasound (EUS), he was found to have features of CP. He was transferred to the CP group, which had originally recruited 10 participants who had remained throughout the duration of the breath collection; therefore, 12 participants with CP were analysed, with a median age of 64 (IQR 52–70) and 10 (83.3%) were male.

### 3.2. Pancreatic Exocrine Insufficiency Results

The medians (IQRs) cPDR_240_ for the PDAC, CP, and HV groups were 14.2% (7.3–28.8), 15.9% (2.2–27.0), and 31.5% (28.1–37.4), respectively (*p* < 0.001). The area under the ROC (AUROC) curve for using cPDR_240_ to discriminate between PDAC and HVs was 0.83 (95% CI: 0.70–0.96) (Figure 1a). For CP versus HV, the AUROC was 0.86 (95% CI: 0.74–0.99) (Figure 1b). Using 18.9% as the defined cut-off for PEI at 240 min [11], PEI was present in 15/23 (65.2%) with PDAC, 6/12 (50.0%) with CP, and 0/24 HVs (0.0%).

A fresh or historical FE-1 result was available for 47 participants: 20/23 (87.0%) in the PDAC group, 7/12 (58.3%) in the CP group, and 20/24 (83.3%) HVs. The medians (IQRs) FE-1 for the PDAC and CP groups were 87.5 µg/g (15–500) and 21 µg/g (15–386), respectively. Every participant in the HV group exhibited FE-1 values above the upper limit of detection (>500). The AUROC curve for using FE-1 to discriminate between PDAC and HVs was 0.85 (95% CI: 0.75–0.95) (Figure 2a) and for CP versus HVs the AUROC was 0.93 (95% CI: 0.78–1.00) (Figure 2b). Using FE-1 < 200 µg/g as the cutoff, PEI was present in 11/20 (55.0%) with PDAC, 5/7 (71.4%) with CP, and 0/20 HVs (0.0%).

PEI was detected by either the ^13^C-MTGBT or FE-1 in 17/23 (73.9%) with PDAC and 8/12 (66.7%) with CP. An attempt to combine estimates from both tests into multivariable logistic regression models with PDAC versus HVs and CP versus HVs as the outcomes could not be conducted for all groups because every FE-1 value in HVs was >500 µg/g. The optimal cutoff point estimated from the maximum of the Youden index for the ROC curve discriminating between PDAC and HVs using cPDR_240_ is 17.7%. For FE-1, this estimated cutoff point is 485 µg/g. The maximum of the Youden index for the ROC curve discriminating between CP and HVs using cPDR_240_ is 27.1%. For FE-1, this estimated cutoff point is 386 µg/g. These calculations are demonstrated in Appendix A.

### 3.3. The Impact of New-Onset Diabetes Mellitus

Seven of the twenty-three (30.4%) participants with PDAC were diagnosed with NODM. Using Fisher’s exact test, the prevalence of NODM in those with PDAC was significantly higher compared with the other groups. Within the PDAC group, the median (IQR) cPDR_240_ for the NODM vs. no NODM groups was 11.5% (6.3 – 26.7) and 14.9% (7.6–29.5), respectively (*p* = 0.53). The median (IQR) FE-1 values for the NODM vs. no NODM groups were 257.5 µg/g (15–500) and 87.5 µg/g (17–485), respectively (*p* = 1.00). One of the twelve (8.3%) participants within the CP group was diagnosed with NODM. 

## 4. Discussion

The novel results of our pilot study show that PEI occurs in resectable PDAC to a similar extent as in CP. We have demonstrated that the ^13^C-MTGBT and FE-1 tests can discriminate between resectable PDAC and HV. These data also show that PEI is a feature of early PDAC, when it is still amenable to surgical resection. This implies that PEI could be used as a proxy to facilitate the early detection of PDAC using either the ^13^C-MTGBT or FE-1 test. Both possess good discriminatory abilities for identifying PEI, which occurs in resectable PDAC to a similar extent as in CP when compared with HVs. The optimal performance of the ^13^C-MTGBT to discriminate between PDAC and HVs occurs at a percentage cPDR_240_ that is closer to the validated threshold for diagnosing PEI than the optimal performance of the FE-1 test. In contrast, the optimal performance of the FE-1 test to discriminate between CP and HVs occurs at a concentration close to the minimum threshold for the FE-1 test in routine practice (<15 µg/g). The ^13^C-MTGBT may be more suitable for detecting PEI due to PDAC, and the FE-1 test could be more appropriate for CP. 

A systematic review and meta-analysis by Iglesia et al. found the pooled prevalence of PEI in advanced PDAC to be 72% (95% CI: 55–86%) from seven separate studies [21]. The diagnoses of PEI in this systematic review were made from a range of diagnostic criteria, including FE-1 < 200 µg/g and a precursor of the ^13^C-MTGBT—the ^13^C-triolein breath test [22]. This is slightly higher than the prevalence of PEI in resectable PDAC seen in our study detected by the ^13^C-MTGBT (65.2%) and FE-1 (55.0%), which may be explained by more advanced disease correlating with worsening pancreatic exocrine function. A separate systematic review and meta-analysis by Powell-Brett et al. on the effectiveness of the ^13^C-MTGBT for PEI found a pooled sensitivity of 0.84 (95% CI: 0.73–0.91) and specificity of 0.87 (95% CI: 0.79–0.93) [10]. These were pooled from six studies that had larger test populations (except one [23]) and longer collection times [9,11,24,25,26]. They reviewed a range of collection times from four to nine hours after consumption of the test meal, with varying amounts of ^13^C-MTG and utilised unlabelled fat. The 240 min-window used to collect the breath samples in our study was shorter than many of the other ^13^C-MTGBT studies featured in this systematic review; however, we decided, for the comfort of the participants, that there was sufficient evidence to test over this duration [11,27]. Additionally, there was some discussion about the quantity of unlabelled fat that should be consumed with the meal to stimulate lipase production by the pancreas. Our study demonstrates that the addition of 10 g of unlabelled fat to the test meal is sufficient to detect a difference in ^13^CO_2_/^12^CO_2_ at 240 min as opposed to using a greater quantity, as is sometimes recommended [28].

Our test confirms that 240 min is an appropriate time frame to collect breath samples. However, 240 min is still a lengthy collection period, which is a limitation of the breath test for its implementation in routine clinical practice. Another limitation of using the ^13^C-MTGBT includes the availability of locations that are able to process and evaluate the samples. The stability of ^13^CO_2_ enables the breath samples to be transferred to other locations without specific requirements such as temperature control; however, having to transport samples between institutions would be associated with additional costs. One strategy could address both limitations and use the stability of the ^13^C isotope to enable these breath tests to be conducted in the comfort of participants’ homes, with samples posted to a central laboratory. This approach is likely to be cheaper and more convenient than a hospital visit [29,30]. Variations in the preparation of the test meal with ^13^C-MTG powder were also a potential limitation given the small quantities of triglyceride required (250 mg), which might be more challenging to prepare in a home environment. This could be circumvented using capsules containing precise quantities of the triglyceride mixture, but this mechanism of delivery requires a feasibility study to determine the efficacy of delivering the test meal in this manner. The biggest limitation with FE-1 testing is that samples may not be returned. In bowel cancer screening programs, failure to return samples was often attributed to forgetting about the kit or hygiene concerns [31,32]. More than 80% of the stool samples were returned in those with PDAC and HVs in this study, but most of these samples were made available during the 240 min in which test was conducted, and this was clearly explained on the participant information sheet. If transferred into a community setting, it is anticipated that the return rates might be lower. The number of facilities able to process these stool samples for FE-1 is also limited; however, it is also a stable measure and is suitable for transportation. 

In addition to demonstrating the potential utility for early detection, this study provides further evidence that PEI is often a feature of resectable PDAC. This highlights the importance of ensuring that these individuals are prescribed PERT, as it is associated with increased survival in PDAC [33] but it is often underappreciated, and subsequent prescription rates are often poor at around 20% of patients [34]. This is particularly important as malnutrition is independently associated with post-operative morbidity and mortality after pancreatic surgery [35].

Populations who would benefit the most from undertaking these PEI tests are those at high risk of developing PDAC. This includes those with a significant family history of PDAC or in those with NODM [36]. These tests could be used as complementary investigations following an NODM diagnosis with HbA1c, given the association between dysglycaemia detected by HbA1c and incident PDAC [37]. Due to the prevalence of PDAC, these tests would probably be most cost-effective if applied to a cohort of patients with NODM rather than to the general population. The ^13^C-MTGBT costs approximately GBP 75 per individual at our institution, with FE-1 costing GBP 39 per sample. 

The prevalence of NODM in those with PDAC compared with the other groups was another significant finding from this study; however, the presence of NODM did not have a significant impact on either test for PEI. The utility for detecting PEI as a proxy for PDAC in those with NODM requires further evaluation with a study in a large cohort of NODM participants with and without PDAC to determine whether either test is a suitable screening tool to facilitate the early diagnosis of PDAC among this population. Ideally, this would then be applied in a prospective setting amongst populations with NODM. Either test could be applied as a single screening test, but may be of greater benefit if repeated measurements with the ^13^C-MTGBT or FE-1 tests could be undertaken as part of a surveillance strategy investigating declining exocrine function as part of a longitudinal study. In the event that ^13^C-MTGBT or FE-1 tests are applied to screen for PDAC, they should only be used to determine if further investigations are warranted in the event that PEI is detected to avoid a false positive test due to CP. Until further studies are performed, the current recommended thresholds of 18.9% for the ^13^C-MTGBT and <200 µg/g for FE-1 should be used to prompt further evaluation with either a CT scan or EUS, depending on local policy.

This pilot study has demonstrated PEI to be a feature of resectable PDAC. Power calculations were conducted to determine the number of participants required to show a difference between the PDAC and HV populations, although we acknowledge that further studies across multiple sites using larger sample sizes are necessary. There were 25 planned individuals with PDAC for this study, but even with 23 included in the analysis, this still resulted in >99% power. This would increase the diversity of the participant pool and enable more accurate analysis of demographics, which might influence the outcome, such as different ethnic or socio-economic backgrounds. This would enhance the reliability and applicability of the findings. Our study was conducted in participants who did not have any evidence of metastasis, as PDAC commonly spreads to the liver (60%), lungs and peritoneum (30%), and bone (10%), primarily to determine which participants would be suitable for surgical intervention as metastases are currently a contra-indication to resection in our centre. We also avoided participants with metastatic disease as liver metastases may affect the hepatic function and subsequent conversion of the ^13^C-MTG to ^13^CO_2_; lung metastases may impact the respiratory function and expiration of ^13^CO_2_; and bone metastases might induce a state of hypercalcaemia that could affect pancreatic function [38].

Future directions for these tests should involve prospective studies performed in larger population groups with sufficient numbers to enable subgroup analysis to incorporate recognised risk factors for developing PDAC. This includes those with NODM as well as those with a strong family history of PDAC. They should be conducted with repeated visits to a testing facility to determine the coefficient of variation between test subjects at different time points within a short period of days to weeks, with longitudinal measurements over a longer timeframe, e.g., annually, to measure if there is any deterioration in exocrine pancreatic function, which could be suggestive of an underlying issue that would then require further investigation.

## 5. Conclusions

PEI occurs in resectable PDAC to a similar extent as in CP, and further work is needed to test whether the measurement of PEI could be used as an initial test to identify people at risk of PDAC in primary care settings. The optimal performance of the ^13^C-MTGBT to discriminate between PDAC and controls occurs at a percentage cPDR_240_ (17.7%), similar to the threshold for diagnosing PEI (<18.9%). The ^13^C-MTGBT could be a good alternative assessment for identifying PEI in resectable PDAC compared with the FE-1 test. Further prospective large-scale studies are warranted in populations at high risk of developing PDAC, such as those with NODM, to facilitate a change in practice and ensure the earlier detection of PDAC.

## Figures and Tables

**Figure 1 cancers-15-05756-f001:**
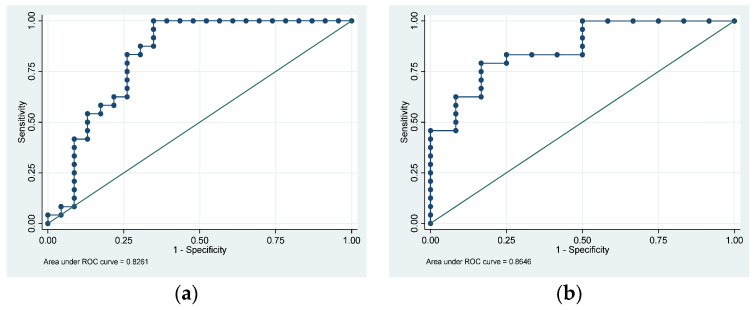
Receiver operating characteristic curves using the 240-min cumulative percentage dose recovered of ^13^CO_2_ compared with ^12^CO_2_ (cPDR_240_) to discriminate between (**a**) pancreatic ductal adenocarcinoma and healthy volunteers and (**b**) chronic pancreatitis and healthy volunteers.

**Figure 2 cancers-15-05756-f002:**
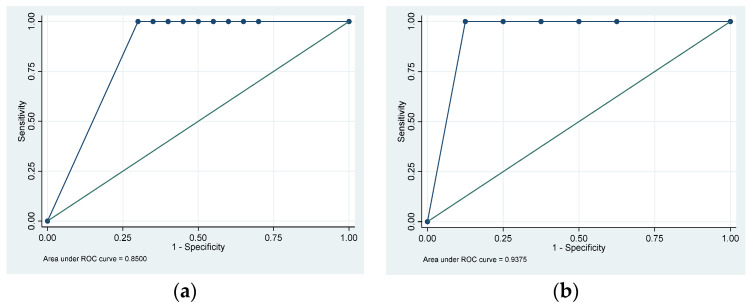
Receiver operating characteristic curves using Faecal Elastase-1 to discriminate between (**a**) pancreatic ductal adenocarcinoma and healthy volunteers and (**b**) chronic pancreatitis and healthy volunteers.

**Table 1 cancers-15-05756-t001:** Study participant characteristics and results from testing for pancreatic exocrine insufficiency.

Participant Characteristics	Healthy Volunteers (*n* = 24)	Pancreatic Ductal Adenocarcinoma (*n* = 23)	Chronic Pancreatitis (*n* = 12)	*p* Value
Age, Years *,	63 (58–71)	68 (56–75)	64 (52–70)	0.63
Sex ^†^, Men (%)	13 (54.2%)	16 (69.9%)	10 (83.3%)	0.17
Weight, kg ^#^	81.3 ± 19.9	77.1 ± 9.6	79.9 ± 20.3	0.70
Body Mass Index, kg/m^2 #^	28.3 ± 6.5	26.0 ± 3.7	26.7 ± 6.0	0.36
Diabetes Mellitus ^†^, Yes	2 (8.3%) ^a,b^	8 (34.8%)	4 (33.3%)	0.09
New-Onset Diabetes Mellitus ^†^, Yes (<36 months since diagnosis of diabetes mellitus)	0	7 (30.4%)	1 (8.3%)	0.004
Cumulative Percentage Dose Recovered of ^13^CO_2_ at 240 min, % *	31.5 (28.1–37.4) ^a,b^	14.2 (7.3–28.8)	15.9 (2.2–27.0)	<0.001
Faecal Elastase-1, µg/g *	500 (n/a) ^a,b^	87.5 (15–500)	21 (15–386)	<0.001

Data are presented as means ± SD ^#^ or medians (IQR) * for normally and non-normally distributed variables, respectively. Variables with dichotomised outcomes are labelled with ^†^. ^a^ Significant difference (*p* < 0.05) detected between healthy volunteers and pancreatic ductal adenocarcinoma groups. ^b^ Significant difference (*p* < 0.05) detected between healthy volunteers and chronic pancreatitis groups.

## Data Availability

The data that support the findings of this study are available from the corresponding author, D.M., upon reasonable request.

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
