# Peer review of "Utilising Pancreatic Exocrine Insufficiency in the Detection of Resectable Pancreatic Ductal Adenocarcinoma"

_cancers, 2023, doi:10.3390/cancers15245756_

Round 1

Reviewer 1 Report

Comments and Suggestions for Authors

I appreciate the authors effort to identify potential methods for early detection of PDAC.  It is a deadly disease with only cure of surgery at early stages.  The study was well conducted and I have no significant issue with the findings and conclusions.

I would like the authors to include additional paragraphs in the discussion regarding potential economic costs when the test is applied only to the NODM cases and to the general public.  Also, at what threshold would the tests trigger additional clinical workup such as either CT or endoscopy.  

Author Response

Response: Thank you taking the time to review the manuscript and for these helpful comments. We have subsequently revised the discussion to incorporate your suggestions. We have emphasised that the test would be most cost-effective in a NODM population compared to the general population, with the test costing approximately £75 and £39 for 13C-MTGBT and FE-1, respectively. We think the value of a positive test would be to trigger a further evaluation of the pancreas with either a CT scan or EUS, depending on local practice. We didn’t include a health economic modelling component in this particular study but future larger studies would evaluate the cost per Quality adjusted life year (QALY) which would be required for NICE to approve for routine Standard of Care. We have also acknowledged and emphasised that the current recognised thresholds should be employed to trigger additional clinical workup until larger studies have been conducted. These revision can be found in lines 317-335.

Reviewer 2 Report

Comments and Suggestions for Authors

In the study's abstract, the authors discuss Pancreatic Ductal Adenocarcinoma (PDAC), typically diagnosed at a late stage, leading to high mortality rates. They aimed to determine if two tests for PEI could differentiate between patients with operable PDAC and healthy individuals, as well as those with chronic pancreatitis (CP). The study involved 59 participants, including those with PDAC, healthy volunteers, and individuals with CP, all of a similar age range. The results showed that the tests could effectively distinguish PDAC from healthy individuals, with accuracy similar to differentiating CP from healthy individuals. These findings underscore the presence of PEI in resectable PDAC, akin to its presence in CP, and suggest the potential of these tests in early PDAC detection, particularly in high-risk groups in a primary care setting.

he study described employed a range of methods to assess pancreatic exocrine insufficiency (PEI) in participants. These methods included the 13C-mixed triglyceride breath test (13C-MTGBT) and a fecal elastase test (FE-1). The participants, including those with Pancreatic Ductal Adenocarcinoma (PDAC), chronic pancreatitis (CP), and healthy volunteers (HV), were tested at the National Institute for Health and Care Research Southampton Clinical Research Facility. They followed specific protocols like fasting, refraining from pancreatic enzyme replacement therapy, and providing stool samples. The breath test involved collecting breath samples before and after consuming a test meal and then analyzing these samples for carbon isotopes.

For statistical analysis, the study used the Skewness-Kurtosis method for normality testing, Kruskal-Wallis test for assessing differences in test results among groups, and receiver operator characteristic (ROC) curves for discrimination between disease states and healthy volunteers.

Limitations and Suggestions:

    1. Sample Size and Diversity: The study's sample size and composition might limit the generalizability of the results. Including only 59 participants and basing the sample size calculation on previous studies might not provide a comprehensive view of the population. Increasing the sample size and including a more diverse range of participants in terms of demographics and disease stages could enhance the reliability and applicability of the findings.

    2. Single Location Study: Conducting the study in a single clinical research facility might introduce location-specific biases. Expanding the study to multiple centers would increase the validity of the results across different populations and healthcare settings.

    3. Methodological Variability: The reliance on participant compliance for fasting and refraining from pancreatic enzyme replacement therapy might introduce variability in the results. Standardizing these conditions more rigorously or exploring methods that are less dependent on participant compliance could improve the consistency of the data.

    4. Long-term Follow-Up: The study lacks long-term follow-up data. Including follow-up assessments could provide insights into the progression of the disease and the long-term efficacy of the testing methods used.

    5. Comparative Analysis with Other Diagnostic Methods: While the study focuses on 13C-MTGBT and FE-1 tests, comparing these with other diagnostic methods could provide a clearer picture of their relative effectiveness and accuracy.

    6. Detailed Analysis of Subgroups: More detailed analysis of subgroups, such as different stages of PDAC or variations in CP, would offer a deeper understanding of how these conditions affect test results.

    7. Exploration of Confounding Factors: The study could benefit from a more thorough investigation of potential confounding factors, such as lifestyle variables or concurrent medical conditions, which might influence the results of the tests.

    8. Furthermore,
      1. Progression and Metastasis in PDAC: It's crucial to recognize that PDAC is a highly aggressive cancer, often diagnosed at a late stage. While the discussion primarily focuses on PEI in resectable PDAC and its early detection, it's essential to acknowledge the potential for disease progression. In advanced stages, PDAC can metastasize to distant organs, including bones.

      2. Bone Metastasis and its Consequences: When discussing the management and detection of PEAC in PDAC, consider that bone metastasis in PDAC, though less common than in other cancers, can lead to severe complications like bone pain, fractures, and hypercalcemia. These complications can significantly impact patient quality of life and treatment outcomes.

      3. Implications for Early Detection and Treatment: The potential for bone metastasis in advanced PDAC underscores the importance of early detection strategies, like the 13C-MTGBT and FE-1 tests. Early detection and treatment of PDAC could potentially reduce the risk of metastasis, including to bones.

      4. Integrated Approach to Patient Care: In the broader context of PDAC management, the discussion could benefit from emphasizing the need for an integrated approach to patient care. This approach would not only focus on early detection of PEI but also on monitoring for signs of metastasis, including to the bones, especially in patients with advanced disease.

      5. Research Gaps and Future Directions: Finally, the discussion can be expanded to highlight the need for further research into the metastatic pathways of PDAC, including the incidence and impact of bone metastasis. This research could lead to better understanding and management of PDAC in its advanced stages.

By integrating these aspects, the discussion would present a more comprehensive view of PDAC, acknowledging its potential for systemic effects like bone metastasis, and emphasizing the importance of early detection and an integrated approach to patient care. Indeed bone metastases, pancreatic ductal adenocarcinoma, PDAC are crucial issues to be discussed.

Incorporating the concept of bone metastases into the authors' discussion on pancreatic exocrine insufficiency in PDAC can be approached by considering the broader implications of PDAC progression.

Comments on the Quality of English Language

In the study's abstract, the authors discuss Pancreatic Ductal Adenocarcinoma (PDAC), typically diagnosed at a late stage, leading to high mortality rates. They aimed to determine if two tests for PEI could differentiate between patients with operable PDAC and healthy individuals, as well as those with chronic pancreatitis (CP). The study involved 59 participants, including those with PDAC, healthy volunteers, and individuals with CP, all of a similar age range. The results showed that the tests could effectively distinguish PDAC from healthy individuals, with accuracy similar to differentiating CP from healthy individuals. These findings underscore the presence of PEI in resectable PDAC, akin to its presence in CP, and suggest the potential of these tests in early PDAC detection, particularly in high-risk groups in a primary care setting.

Author Response

Response: Thank you taking the time to review the manuscript and for these helpful comments. We have subsequently revised the discussion to incorporate your suggestions. We acknowledge in the manuscript that there are limitations with conducting a study of this size in a single location, however this was a feasibility study to act as a proof-of-concept, to determine if these tests can be conducted on a larger scale trial before commencing enrolment across multiple centres with a larger, more diverse study population. The study is closed to recruitment now, and it was funded by CRUK with the intention of being a pilot study. Sample size calculations were conducted prior to enrolment in the test as a requirement of the funder, which were adequate to demonstrate a significant difference in this pilot study (published as Afolabi PR, McDonnell D, Byrne CD, Wilding S, Goss V, Walters J, Hamady ZZ. DEPEND study protocol: early detection of patients with pancreatic cancer - a pilot study to evaluate the utility of faecal elastase-1 and 13C-mixed triglyceride breath test as screening tools in high-risk individuals. BMJ Open. 2022 Feb 25;12(2):e057271. doi: 10.1136/bmjopen-2021-057271. PMID: 35217541; PMCID: PMC8883257) and were deemed adequate by the Cancer Research UK funding committee (which included statisticians).   

The original power calculation was undertaken using data collected from a previous study (Kato H, Nakao A, Kishimoto W, et al. 13C-labeled trioctanoin breath test for exocrine pancreatic function test in patients after pancreatoduodenectomy. Am J Gastroenterol 1993;88:64–9.pmid:http://www.ncbi.nlm.nih.gov/pubmed/8093586). The power calculation was for a two-sided test comparing the mean (standard deviation) of the cumulative percentage dose recovered of 13CO2 at 240 minutes. For the healthy volunteers, their data showed a mean of 42% (S.D. 3.4) versus 14.2% (S.D. 10.5) for the PDAC group. Originally 25 individuals in the PDAC and healthy volunteer groups were planned to be recruited, which would attain a power of >99%. In the analysis, 23 with PDAC and 25 volunteers were included. The expected power can then be calculated:

power twomeans 42 24.2, sd1(3.4) sd2(10.5) n1(25) n2(23)

This gave an estimated power of >99%, the same value as the original planned analysis of 25 per group.

We acknowledge relying on participants fasting and refraining from pancreatic enzyme replacement therapy (PERT) might introduce variability, however we often had a very short window of opportunity between diagnosis and intervention to undertake the study. We think participants would have been unwilling to be admitted the night before to ensure they were all sufficiently fasted and not taking their medications. The PDAC group would be at significant risk of malnutrition and therefore the Ethics committee may not have approved prolonged periods of pre-test fasting and withholding of PERT. Having demonstrated these tests can work as a pilot study, more rigorous methods could be used with further tests. We will take these suggestions into consideration with future studies of this manner. We explored the long-term impact of those with and without PEI in the PDAC group following surgical intervention, however the dataset was too small to draw meaningful conclusions. If applied to a larger sample population, more meaningful conclusions could be drawn; but this study was designed to demonstrate if PEI determined by the 13C-MTGBT or FE-1 was a feature of resectable PDAC. Ideally, our study would have also included the gold standard test of the coefficient of fat absorption test in order to generate sensitivities and specificities, however we have referenced that both tests provide similar results to this coefficient of fat absorption test in the manuscript, and our aim was to test if the 13C-MTGBT and faecal elastase can be used to discriminate between PEI and normality in PDAC, compared to healthy volunteers. In addition, most work on PEI is conducted by FE and/or 13C-MTGBT. Rarely is fat coefficient measured due to its cumbersome nature and there are many papers testing the reliability of both FE and 13C-MTGBT. Additionally, we considered that this was too much to ask of our patients who were going through a very traumatic period in their lives. We have also expanded the manuscript to look at participants with new onset diabetes mellitus (NODM), but as this study has been powered to determine if PEI is a feature of resectable PDAC compared to healthy volunteers, there was no discernible difference in PEI in those with NODM compared to those without NODM in the PDAC group. If conducted on a larger population, specifically in those with and without NODM, more robust conclusions might be made as we would be able to stratify by patient populations as part of the study design. It would also require a larger study to explore the impact of other confounding factors as the sample size is too small in this study to have sufficient statistical power. We have expanded upon these issues in lines 336-341 of the discussion.

Our study aim was to focus on those with resectable PDAC i.e. without metastases, and to test if 13C-MTGBT and faecal elastase can be used to discriminate between PEI and normality in PDAC. We have acknowledged the importance of metastatic disease in more detail at the end of the manuscript with specific focus on liver, lung and bone metastases and explained why these might confound the results if these populations were included. Our focus was on identifying persons who would be suitable for surgical resection, rather than diagnosis of all stages of PDAC. If conducted on a larger population in a prospective manner, it is highly likely the majority would have had metastasis given the nature of the disease. We acknowledge this important aspect highlighted by the reviewer and thank them for raising this important point. We have discussed it in lines 344 – 351.

Reviewer 3 Report

Comments and Suggestions for Authors

In this study, McDonnell et al. aim to determine whether two recognized tests for pancreatic exocrine insufficiency (PEI) - the 13C-mixed triglyceride breath test (13C-MTGBT) and the fecal elastase (FE-1) test - could differentiate between individuals with resectable pancreatic ductal adenocarcinoma (PDAC) and healthy subjects. The findings revealed that PEI is as prevalent in resectable PDAC as it is in chronic pancreatitis (CP). Both the 13C-MTGBT and FE-1 tests showed strong discriminatory capabilities in identifying PEI, a characteristic of early PDAC. The authors concluded that larger-scale studies are necessary to validate the potential use of these tests for early PDAC detection. However, several issues need to be addressed for the acceptance of the article:

  1. The sample size of the study is relatively small, comprising only 59 participants (23 with PDAC, 24 healthy volunteers, and 12 with CP). This limits the statistical power of the study, and the authors should include more participants to strengthen the conclusions.
  2. The study was conducted at a single location in the UK, which could limit the diversity of the participant pool. Along with the inclusion of more participants, they should ensure that they encompass populations with different ethnic, socio-economic, or geographical backgrounds. 
  3. The authors suggest that the tests could be beneficial for high-risk groups, such as those with new-onset diabetes mellitus (NODM), but these populations are not included in the study. 
  4. The study should be longitudinal to provide insights into how PEI or PDAC progresses over time in the same individuals.

Author Response

Response:  Thank you taking the time to review the manuscript and for these helpful comments. We have subsequently revised the discussion to incorporate your suggestions. We acknowledge in the manuscript that there are limitations with conducting a study of this size in a single location, however this was a feasibility study to act as a proof-of-concept, to determine if these tests can be conducted on a larger scale trial before commencing enrolment across multiple centres with a larger, more diverse study population. The study is closed to recruitment now, and it was funded by CRUK with the intention of being a pilot study. Sample size calculations were conducted prior to enrolment in the test as a requirement of the funder, which were adequate to demonstrate a significant difference in this pilot study (published as Afolabi PR, McDonnell D, Byrne CD, Wilding S, Goss V, Walters J, Hamady ZZ. DEPEND study protocol: early detection of patients with pancreatic cancer - a pilot study to evaluate the utility of faecal elastase-1 and 13C-mixed triglyceride breath test as screening tools in high-risk individuals. BMJ Open. 2022 Feb 25;12(2):e057271. doi: 10.1136/bmjopen-2021-057271. PMID: 35217541; PMCID: PMC8883257. )

The original power calculation was undertaken using data collected from a previous study (Kato H, Nakao A, Kishimoto W, et al. 13C-labeled trioctanoin breath test for exocrine pancreatic function test in patients after pancreatoduodenectomy. Am J Gastroenterol 1993;88:64–9.pmid:http://www.ncbi.nlm.nih.gov/pubmed/8093586). The power calculation was for a two-sided test comparing the mean (standard deviation) of the cumulative percentage dose recovered of 13CO2 at 240 minutes. For the healthy volunteers, their data showed a mean of 42% (S.D. 3.4) versus 14.2% (S.D. 10.5) for the PDAC group. Originally 25 individuals in the PDAC and healthy volunteer groups were planned to be recruited, which would attain a power of >99%. In the analysis, 23 with PDAC and 25 volunteers were included. The expected power can then be calculated:

power twomeans 42 24.2, sd1(3.4) sd2(10.5) n1(25) n2(23)

This gave an estimated power of >99%, the same value as the original planned analysis of 25 per group.

We have expanded upon these issues in lines 336-344 of the discussion.

We have also stratified our data by those with new onset diabetes mellitus (NODM). The prevalence of NODM is higher in PDAC than in the other groups, but this does not seem to affect either test for PEI. We emphasise towards the end of the discussion that this would require a separate, larger study investigating NODM in those with and without PDAC, to determine what the impact would be. These changes were incorporated in lines 87-90, 162-163, table 1, 252-260, 321-323.

We also acknowledge that repeated measurements conducted in a longitudinal manner would be useful as surveillance to detect potential development of early PEI. This would require a much larger sample size but may be feasible if targeting subjects with a diagnosis of NODM within a prospective cohort study design. The parameters of this study were also configured for a single attendance to determine if the 13C-MTGBT or FE-1 could discriminate between PDAC and HV as a pilot test as opposed to repeat testing of their measurements over time. The PDAC cases in this test underwent surgical resection shortly after attending this study as this was imperative for the clinical care of the patients with this devastating disease that has such a poor prognosis (and area of unmet need for Cancer Research UK).. This limited evaluation on this population in this study.  We have discussed in lines 326-330.

Of the 23 participants with PDAC, 12 went ahead with surgical resection. The reasons for not having surgery included other co-morbidities precluding major surgical resection, being unable to undertake pre-operative fitness evaluation, complications arising from interventions such as stenting of the PDAC resulting in pancreatitis and patient choice not to proceed to surgery. Due to the aggressive nature of this cancer, those that underwent surgical intervention had significantly higher one year survival than those who did not. However, due to the low numbers involved and the fact they attended the study once, we were unable to draw any meaningful conclusions regarding PEI progression or PDAC progression in the small number of survivors from this devastating life limiting cancer.

Round 2

Reviewer 2 Report

Comments and Suggestions for Authors

I am satisfied.

Comments on the Quality of English Language

I am satisfied.

Author Response

Many thanks for taking the time to review the manuscript and providing constructive comments

Reviewer 3 Report

Comments and Suggestions for Authors

The authors have responded to every concern raised in the previous report. Their rebuttal and limitations are understandable. The authors are recommeded to add a more detailed limitations and future directions section.

Author Response

Thank you for taking the time to review this manuscript and provide constructive comments. We have added additional paragraphs focusing on limitations and expanded upon our previous manuscript. We have also detailed future directions at the end of the discussion